# Obstructive Tracheobronchitis in Influenza-Associated Pulmonary Aspergillosis

**DOI:** 10.3390/diagnostics14151628

**Published:** 2024-07-28

**Authors:** Shuyi Zhang, Chen Shang, Zhangjun Tan, Wen Li

**Affiliations:** Department of Intensive Care Medicine, Renji Hospital Affiliated to Shanghai Jiao Tong University School of Medicine, No. 1630 DongFang Road, Pudong New District, Shanghai 200127, China; shangchen6558@renji.com (C.S.);

**Keywords:** obstructive tracheobronchitis, invasive pulmonary aspergillosis, bronchoscopy

## Abstract

We report a bronchoscopic image of a 36-year-old with significant airway obstruction from obstructive tracheobronchitis secondary to invasive pulmonary aspergillosis. It is rare to see such a severe form of obstructive tracheobronchitis, likely caused by the patient’sp immunocompromised status and rapid progression nature of influenza-associated pulmonary aspergillosis.

A 36-year-old man presented to the Emergency Department (ED) with a one-week history of cough and fever. He had a past medical history of kidney transplant ten months prior and was on long-term tacrolimus. He tested positive for influenza B. A CT scan revealed only bronchitis, with no parenchymal lung changes (Appendix A). He was discharged home with oseltamivir and symptomatic treatment. 

He returned to ED three days later with severe respiratory distress and was immediately transferred to the ICU and intubated due to hypoxia and increased work of breathing. Following intubation, he was hemodynamically stable. Physical examination revealed significantly decreased breath sounds bilaterally. Ventilator waveforms indicated high airway pressure and increased resistance. An urgent bronchoscopy was performed, revealing significant (>75%) narrowing in the main bronchus. Immediate debulking with biopsy forceps in the main bronchus was performed to assist ventilation. Bilateral main bronchi exhibited obstructive tracheobronchitis with prominent mucous plugs (Figure 1) and pseudomembrane (Figure 2). A bedside chest X-ray following intubation showed mild ground-glass opacification in the right lung (Figure 3). The CT scan performed the day after intubation and ICU admission revealed new bilateral patchy opacities and small nodular lesions in the lung parenchyma (Appendix A).

Bronchoalveolar lavage fluid (BALF) galactomannan (GM) was 10.86 DOI (*Aspergillus* Galactomannan Detection Kit (CLIA); Genobio Pharmaceutical Co., Ltd.). BALF culture grew *Aspergillus fumigatus*. The diagnosis of invasive tracheobronchial aspergillosis (ITBA) secondary to probable influenza-associated pulmonary aspergillosis (IAPA) was made [1]. According to the proposed case definition of IAPA in ICU patients, in patients with an influenza-positive diagnostic test, influenza-like illness, and bronchoscopic evidence of tracheobronchitis, such as our patient, a positive BAL GM or culture of a tracheal aspirate is considered mycological evidence supporting a probable IAPA diagnosis [2].

The patient was treated with intravenous voriconazole and nebulized amphotericin B. Tacrolimus was discontinued. Daily bronchoscopy was initially required to clear copious purulent thick secretion. After two weeks of treatment, repeated bronchoscopy showed significant improvement in airway obstruction (Appendix A). He required tracheostomy and prolonged ventilation for up to four weeks, but eventually was weaned off the ventilator and transferred from ICU to the general ward. 

This case demonstrated invasive *Aspergillus* tracheobronchitis in severe obstructive and pseudomembranous forms as classified by bronchoscopy. Invasive tracheobronchial aspergillosis (ITBA) is a very aggressive form of invasive pulmonary aspergillosis (IPA) with mortality rate exceeding 80% in some series [3]. While influenza-associated pulmonary aspergillosis (IAPA) has a prevalence of 19.2% in critically ill patients with influenza [4], the reported incidence of ITBA in influenza-associated IPA ranges from 27% [5] to 55% [4]. CT findings of ITBA can be normal, especially in the early stages of the disease. In our patient, a CT scan done just three days before bronchoscopy only suggested bronchitis, highlighting the rapid progression nature of influenza-associated pulmonary aspergillosis in the setting of pre-existing immunosuppression (on long-term tacrolimus). 

Central airway management for obstructive tracheobronchitis typically involves procedures such as debulking with biopsy forceps, toileting with grasping forceps, balloon dilation, and stent insertion, often requiring repeated endoscopic treatments [6,7]. In this case, only one debulking procedure was performed for the severe obstruction. This underscores that antifungal therapy remains pivotal in the treatment of invasive pulmonary aspergillosis. Therapeutic drug monitoring to ensure adequate dosing and frequent bronchoscopy for secretion removal may enhance antifungal penetration. 

## Figures and Tables

**Figure 1 diagnostics-14-01628-f001:**
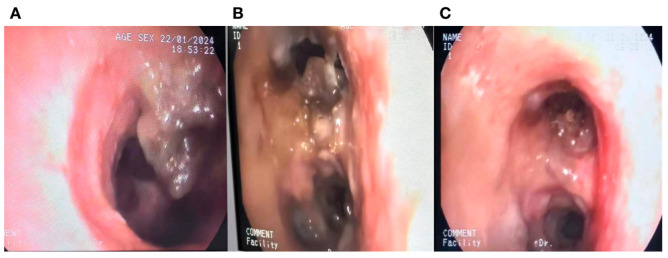
Bronchoscopy performed after intubation in a 36-year-old man with obstructive tracheobronchitis secondary to invasive pulmonary aspergillosis, showing prominent mucous plugs and airway sloughing. (**A**) Left main bronchus, (**B**) Major fissure of left bronchus, (**C**) Right upper Lobe.

**Figure 2 diagnostics-14-01628-f002:**
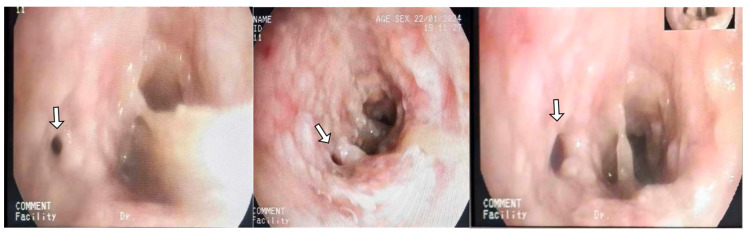
Bronchoscopy of basal segments of the left lower lobe, showing thick pseudomembrane with almost complete closure of anterior basal segment (white arrow).

**Figure 3 diagnostics-14-01628-f003:**
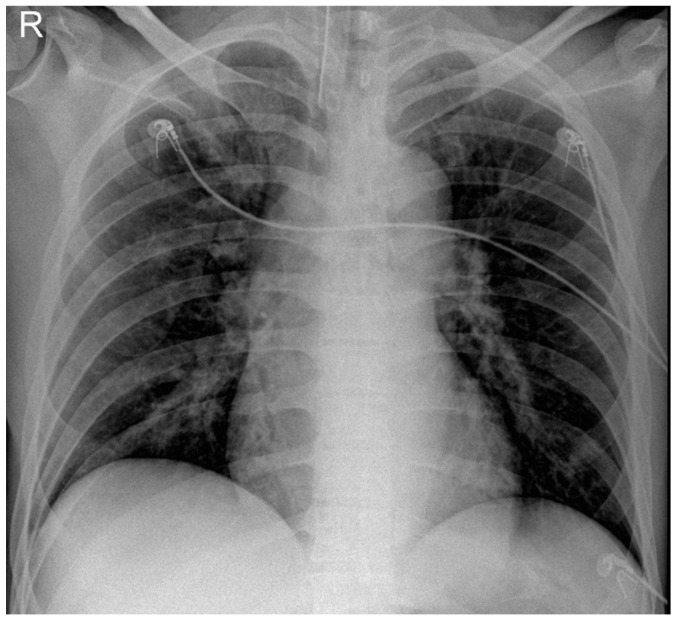
Chest X-ray following intubation showed mild ground-glass opacification in the right (R) lung.

## Data Availability

No new data were created.

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
