# Peer review of "Obstructive Tracheobronchitis in Influenza-Associated Pulmonary Aspergillosis"

_diagnostics, 2024, doi:10.3390/diagnostics14151628_

Round 1

Reviewer 1 Report

Comments and Suggestions for Authors

Zhang and colleagues report a case of obstructive tracheobronchitis in influenza-associated pulmonary aspergillosis.

This case has merit to be published, as it shows the relevance of performing bronchoscopy in patients with severe influenza admitted to ICU.

I do have several remarks that should be addressed:

-          The introduction (lines 16 to 22) is a bit confusing. The way it is written now, I thought the CT scan was taken at the day the patient was intubated, but this was not the case. Please start with the first ED presentation (with chest CT), and only mention the ED presentation where the patient was intubated after describing this first presentation.

-          Was any imaging performed the day the patient was intubated? The authors now only report the CT scan result of the first ED presentation, but no imaging (for instance chest X ray or ultrasound) at time of intubation. 

-          How was the BALF galactomannan assessed, with which assay? Normally, an optical density index (ODI) is given, not µg/L.

-          Was a biopsy for histology (with use of fungal staining) performed? If not, this should be mentioned as a limitation.

-          This case of IPA is a case of IAPA (influenza-associated pulmonary aspergillosis). Aspergillosis in influenza is typically a superinfection (influenza as primary infection, and aspergillosis as superinfection). Please adjust the manuscript accordingly in lines 46-49: “In our patient… highlighting the rapidly progressive nature of influenza-associated pulmonary aspergillosis in the setting of pre-existing immunosuppression.” Please change the abstract sentence likewise. Please change title of the manuscript to “Obstructive Tracheobronchitis in Influenza-Associated Pulmonary Aspergillosis”.

-          Please also add a short explanation (two or three sentences) of the incidence and diagnosis of IAPA (use a recent review, several are available).

- Please also refer to the incidence of tracheobronchitis in IAPA described in literature. The paper by Remy Nyga and colleagues (American Journal of Respiratory and Critical Care Medicine 2020) should be referred to at least.

-          Please add the timepoint at which bronchoscopical images shown in figure 1 and 2 was performed. If possible, also add bronchoscopical images later during ICU or hospital stay of the same segments, to show the evolution under antifungal therapy.

Comments on the Quality of English Language

The English language requires minor grammatical checks. Please change “aspergillus” to “Aspergillus”

Reviewer 2 Report

Comments and Suggestions for Authors

Dear authors,

thank you for the opportunity to read and review the paper. It is interesting and well written.

Was a new CT scan performed after worsening of symptoms and intubation? 

Was the tacrolimus suspended during the first days of anti fungal therapy? The patient had hemodynamic instability?

Author Response

Dear reviewer, thank you very much for taking the time to review this manuscript. Please find the detailed responses below and the corresponding revisions highlighted in the re-submitted files.

==========

Comments 1:  Was a new CT scan performed after worsening of symptoms and intubation? 

Response 1:  Yes, we performed a chest CT the day after intubation. The images have been included in the supplemental file (Figure S2) and described in the manuscript (lines 32-34). However, the quality of the images is poor due to artifacts caused by movement.

Comments 2:  Was the tacrolimus suspended during the first days of anti fungal therapy? The patient had hemodynamic instability?

Response 2:  Yes, tacrolimus was suspended on the first day of ICU admission (from the start of anti-fungal therapy).  He’s hemodynamically stable (added in manuscript Line 24).

Round 2

Reviewer 1 Report

Comments and Suggestions for Authors

Dear Authors,

Thank you for adjusting your manuscript to my comments. There is one of your responses with which I do not agree. The widely used case definition of IAPA published by Verweij et al in Intensive Care Medicine in 2020 is quite straightforward and deserves to be briefly mentioned in your discussion. Your patient fits the criteria for probable IAPA according to this definition. Please add one sentence on this to the manuscript. 

Author Response

Comment 1:  The widely used case definition of IAPA published by Verweij et al in Intensive Care Medicine in 2020 is quite straightforward and deserves to be briefly mentioned in your discussion. Your patient fits the criteria for probable IAPA according to this definition. Please add one sentence on this to the manuscript. 

Response 1: Thank you for you suggestion.  We've added the diagnosis of probable IAPA into the manuscript and mentioned the case definition of IAPA by Verweij et al.  (Line 49-54)